# Burnout of Healthcare Workers Amid the COVID-19 Pandemic: A Follow-Up Study

**DOI:** 10.3390/ijerph182111581

**Published:** 2021-11-04

**Authors:** Yoshito Nishimura, Tomoko Miyoshi, Asuka Sato, Kou Hasegawa, Hideharu Hagiya, Yoshinori Kosaki, Fumio Otsuka

**Affiliations:** 1Department of General Medicine, Okayama University Graduate School of Medicine, Dentistry and Pharmaceutical Sciences, Okayama 7008558, Japan; tmiyoshi@md.okayama-u.ac.jp (T.M.); khasegawa@okayama-u.ac.jp (K.H.); hagiya@okayama-u.ac.jp (H.H.); fumiotsu@md.okayama-u.ac.jp (F.O.); 2Department of Medicine, John A. Burns School of Medicine, University of Hawai’i, Honolulu, HI 96822, USA; 3Center for Graduate Medical Education, Okayama University Hospital, Okayama 7008558, Japan; asukasato@okayama-u.ac.jp; 4Center for Education in Medicine and Health Sciences, Okayama University Graduate School of Medicine, Dentistry and Pharmaceutical Sciences, Okayama 7008558, Japan; pj4w35pr@s.okayama-u.ac.jp

**Keywords:** coronavirus disease 2019 (COVID-19), pandemic, burnout, prevention, intention to leave

## Abstract

The coronavirus disease 2019 (COVID-19) pandemic has posed a significant challenge to the modern healthcare system and led to increased burnout among healthcare workers (HCWs). We previously reported that HCWs who engaged in COVID-19 patient care had a significantly higher prevalence of burnout (50.0%) than those who did not in November 2020 (period 1). We performed follow-up surveys in HCWs in a Japanese national university hospital, including basic demographics, whether a participant engaged in care of COVID-19 patients in the past 2 weeks, and the Maslach Burnout Inventory in February 2021 (period 2) and May 2021 (period 3). Periods 1 and 3 were amid the surges of COVID-19 cases, and period 2 was a post-surge period with a comparatively small number of COVID-19 patients requiring hospitalization. Response rates to the surveys were 33/130 (25.4%) in period 1, 36/130 (27.7%) in period 2, and 56/162 (34.6%) in period 3, respectively. While no consistent tendency in the prevalence of burnout based on variables was observed throughout the periods, the prevalence of burnout tends to be higher in periods 1 and 3 in those who engaged in COVID-19 patient care in the last 2 weeks (50.0%, 30.8%, 43.1% in period 1, 2, and 3, respectively). Given the prolonged pandemic causing stigmatization and hatred against HCWs leading to increased prevalence of burnout, high-level interventions and supports are warranted.

## 1. Introduction

Since the coronavirus disease 2019 (COVID-19) emerged, the public health landscape had to go through dramatic changes. Among the struggles to fight against the virus, frontline healthcare workers (HCWs) have had significant stress and hardship since the beginning of the pandemic. As noted by the Director-General of the World Health Organization and other experts, the COVID-19 pandemic has a considerable negative effect on mental health and well-being, by exacerbating existing mental health disorders, limiting access to healthcare, and posing considerable stress and risks of infection [1,2,3,4]. With the prolonged pandemic, HCWs are at increased risk of psychiatric symptoms than the general population, including depression, anxiety, insomnia, or even suicidal ideation [5,6]. In particular, burnout is a common syndrome in HCWs amid the pandemic. While burnout syndrome is not included in the list of mental disorders in the 10th revision of the International Classification of Diseases, a study suggested its association with major mental disorders such as insomnia, depression, anxiety, and post-traumatic symptoms [7]. We previously reported that HCWs who engaged in care of COVID-19 patients had significantly higher burnout rates (50.0%) than those who did not, and those working in the intensive care unit were more likely to have experienced burnout than those in floors [8]. The study was done amid the third surge of the COVID-19 cases in Japan in November 2020. Unfortunately, the country has had to go through two more significant surges of cases since then [9]. Given the uncertainty concerning how long the COVID-19 pandemic may last, HCW burnout, which is a chronic psychological condition with a loss of enthusiasm and personal accomplishment, feelings of physical and mental exhaustion, and depersonalization [10], continues to be a considerable concern from the occupational and public health perspectives.

It needs to be noted that HCWs are vulnerable to burnout at the baseline, with reported prevalence in Japan around 20–30% before the COVID-19 pandemic [11,12]. So far, studies have reported that the prevalence of burnout among HCWs has further increased in various countries during the pandemic [13,14,15,16,17,18,19,20,21,22,23,24]. A recent systematic review and meta-analysis that quantified the psychological symptoms among frontline HCWs during pandemics noted that the prevalence of burnout was 31.8% [25]. In addition, Kok et al. reported in their longitudinal cohort study that HCWs working in intensive care units experienced worsening burnout symptoms during the COVID-19 pandemic compared to the pre-pandemic periods [26]. However, no reports are available if the prevalence of burnout increases over time during the pandemic.

Amid the prolonged fight against the virus, frustration became pervasive among the general public in Japan, which occasionally turned into hatred and discrimination against HCWs [27,28]. Given the stress from the pandemic and COVID-19 patient care themselves as well as the discrimination and violence, it is easily assumed that the prevalence of burnout might have gone up over the period. We aimed to explore the up-to-date prevalence of burnout in Japanese HCWs amid the ongoing COVID-19 pandemic based on a hypothesis that the prevalence may increase over time, following our previous cross-sectional study in a Japanese national university hospital.

## 2. Materials and Methods

### 2.1. Study Design, Setting, and Participants

We conducted three cross-sectional studies in a fashion of anonymous, self-administered voluntary paper or web-based survey. Participants included physicians, nurses, pharmacists, clinical engineers, and physical therapists in Okayama University Hospital (OUH; a tertiary-care Japanese national university hospital with more than 800 beds). Before the initial cross-sectional study that we previously reported [8], we conducted a power analysis for two proportions to estimate a sample size required to detect the significant differences between the prevalence of burnout among those engaged in the COVID-19 patient care in the past 2 weeks compared to those who did not. We defined the prevalence of burnout in those who engaged in COVID-19 care in the period as 70% and those who did not as 20% in the analysis based on the previous studies [12,19,20]. At a 5% level of significance and 80% test power, the sample size needed was estimated as *n* = 15 in each group (*n* = 30 as a whole). Homogenous purposive samplings were used to survey the 130 healthcare professionals who belonged to the OUH on 1 November 2020. The purposive sampling criteria included HCWs who belonged to or took care of the patients in the Department of General Medicine, Emergency Department/Intensive Care Unit (EICU), Center for Graduate Medical Education. All the participants could potentially take care of COVID-19 patients under investigation (PUI) of COVID-19. Completion of the surveys implied the participants’ consent. As noted in the previous study, the survey was developed through consultation with an expert in medical education at OUH and piloting. Survey instructions and instruments were described in Japanese. All participants were invited to complete the surveys from 13–30 November 2020 (period 1), 15–28 February 2021 (period 2), and 18–31 May 2021 (period 3), in Japan Standard Time. Of note, the hospital faced significant surges of COVID-19 cases during periods 1 and 3, and fewer COVID-19 patients were hospitalized in period 2 since Japan was in the post-third surge period in February 2021. We provided no financial incentives for their participation in the survey. The survey items included basic demographics such as gender, job category, affiliation, years of experience, size of household, and a COVID-19 related item (e.g., “Have you engaged in care of patients with COVID-19 or PUI of COVID-19 in the past 2 weeks?”). We did not collect their ages to protect participants’ anonymity.

### 2.2. Measurements

#### The Maslach Burnout Index (MBI)

We measured burnout with the Japanese translation of the Maslach Burnout Inventory–Human Services Survey (MBI-HSS). The instrument was validated to measure burnout in Japanese HCWs by Higashiguchi et al. [29]. It has 22 items with three domains: emotional exhaustion (EE), depersonalization (DP), and personal accomplishment (PA). Each item has a 7-point Likert scale from “never” or 0 to “daily” or 6. Based on a previous study that tried to find the most commonly used raw score cut-off, we defined a 27 or higher EE score and a 10 or higher DP score (the most common cut-off) as burnout [30]. Maslach et al. noted that PA was an independent subscale without correlation with EE and DP subscales. Thus, we did not use low PA scores (33 or lower; the most commonly used cut-off) to define burnout [31].

### 2.3. Statistical Analysis

Data analysis was performed with JMP version 15.1.0 (SAS Institute Inc., Cary, NC, USA). Mann–Whitney U test was chosen to examine differences in the time participants’ MBI scores given its non-normal distribution. We used Fisher’s exact test for associations between categorical variables. We employed logistic regression analyses to evaluate factors associated with burnout. The threshold for significance was defined as *p* < 0.05.

## 3. Results

Response rates to the surveys were 33 out of 130 (25.4%) in period 1, 36 out of 130 (27.7%) in period 2, and 56/162 (34.6%) in period 3, respectively, for the included HCWs. Participants’ demographic characteristics for each survey are summarized in Table 1. Throughout the periods, similar results were observed regarding the mean years of experience of the participants (mean 11.0–11.5 years), gender balances, or household size. Of note, compared to period 1, more participants belonged to EICU in period 2. In period 3, 19/56 (34.0%) participants noted themselves as “others” regarding affiliations. Those who answered “others” in the question included floating nurses working in COVID-19 dedicated units. Participants’ job categories differed between the periods; in periods 1 and 3, 60–70% of the participants were nurses, whereas physicians were dominant (17/36 (47.2%)) in period 2. Over the study periods, more respondents were involved in the COVID-19 patient care. In period 1, 12/33 (36.4%) were engaged in care of COVID-19 patients or COVID-19 PUI in the past 2 weeks, whereas 26/36 (72.2%) in period 2 and 51/56 (91.1%) in period 3 answered that they worked for COVID-19 patients in the designated periods, respectively.

Table 2 summarize respondents’ answers to the MBI and the prevalence of burnout. Although the scores of EE and DP were higher in those who engaged in care of COVID-19 patients or PUI in periods 1 and 3, no statistically significant differences were noted in the measures throughout the periods compared to those who did not engage in the care of COVID-19 patients or PUI in the past 2 weeks. The mean PA scores (higher score suggests a feeling of better self-accomplishment) were higher in those who engaged in COVID-19 patient care than those who did not consistently without statistically significant differences.

Except for period 1 when the prevalence of burnout was significantly higher in those engaged in the care of COVID-19 care in the last 2 weeks compared to those who did not (50.0% vs. 9.5%, *p* = 0.016), no significant changes in the prevalence of burnout was observed in HCWs involved in COVID-19 patient care compared to those did not in period 2 or 3. Of note, however, in period 3 when most of the respondents were engaged in COVID-19 patient care, 22/51 (43.1%) experienced burnout at the time of the survey. No consistent tendency in the prevalence of burnout based on variables was observed throughout the periods (Table 3). However, those engaged in COVID-19 patient care in the last 2 weeks were more likely to experience burnout in periods 1 and 3.

## 4. Discussion

In this follow-up study, we found that burnout was prevalent in 30–50% of HCWs during the periods. While there were no significant increases in the prevalence of burnout in those who were engaged in COVID-19 patients care than those who did not, interestingly, the prevalence of burnout was considerably higher during the nationwide surges of COVID-19 cases (period 1 and 3) than the post-surge period when less COVID-19 patients were hospitalized (period 2). The results suggest that patient overload and burden amid the pandemic might have contributed to the rise in burnout prevalence. Moreover, it poses a question if the current healthcare system would be sustainable given the uncertainty regarding how long the current COVID-19 pandemic may last.

Unlike our previous report, we could not find any variables related to a statistically significant increase in burnout prevalence. However, upon comparing the results from periods 1 and 3, similar tendencies were observed; nurses and those in EICU tended to have higher odds of burnout than their counterparts. While there are mixed results concerning risk factors for burnout during the COVID-19 pandemic, some literature reported that nurses were more vulnerable to burnout than physicians, especially in ICU settings [21,24,26,32]. Given that nurses are always working in the frontline of patient care throughout their shifts, the results were as expected, and hospital administrators may need to focus on nurses to prevent burnout with measures such as occupational mental health surveillance.

Previously, we discussed the problem of burnout from perspectives of promoting universal health coverage (UHC) and resilience [8,33,34]. The importance of UHC has been unchanged amid the pandemic; rather, it has been more critical than ever before to keep sustainable healthcare delivery. As hospitals are overwhelmed by the surge of COVID-19 patient hospitalization and death, HCWs are pressured to ration care due to lack of oxygen, hospital beds, or workforce. Due to the prolonged pandemic, the general public has been frustrated over restrictions and mandates. Unfortunately, the frustration turned into hatred, stigmatization, or attacks against HCWs [27,28,35,36,37,38,39] because some attribute the lack of hospital beds and care rationing to HCWs being lazy. Burnout of HCWs may be the consequence of the various issues discussed above, leading to increased leave of HCWs from hospitals. While HCWs might still need to know how to protect themselves from burnout by mitigating interventions such as counseling, reducing workload, or resilience building, most of the factors causing burnout of HCWs are beyond what an individual can handle. Organizational or policy-level approaches are needed to mentally and physically protect HCWs at this time. Especially, stigmatization and attack against HCWs should never be allowed, and strong interventions at a government level are warranted.

Moral injury is an important issue that arose during the pandemic. Originally, this term was used to describe the psychological symptoms seen in soldiers returning from war. It is defined as psychological distress caused by the fact that one’s actions, or failure to act, are contrary to one’s moral and ethical code [40]. Unlike common mental disorders such as depression, anxiety, or post-traumatic stress disorder (PTSD), moral injury is not categorized as a mental disorder. However, it is a syndrome sharing similarities with PTSD [41], and HCWs are at increased risk of moral injury during the COVID-19 pandemic because they have been forced to ration care amid the surges of the cases. As moral injury may be closely related to PTSD and burnout [42], early interventions such as providing education about the emotional and social challenges during the pandemic as well as reflection on HCWs’ experience would be necessary.

Several limitations to this study need to be noted. First, the study was comprised of three cross-sectional surveys at a single center with small sample sizes. Thus, we may not be able to conclude causal relationships between burnout and the current COVID-19 pandemic. As discussed in our previous report, the organizational climate might be a confounding factor of the prevalence of burnout, which might have affected the results [43]. Second, while we performed multiple surveys over time, we could not perform follow-up surveys for the same individuals. A longitudinal study design still needs to be pursued to clarify the long-term effect of the pandemic. Third, the response rates of surveys were between 25.4–34.6%. Since those suffering from burnout might have opted out to answer the surveys, the true prevalence of burnout among HCWs might be higher than reported. Despite these limitations, our study may be the first to report the burnout prevalence of HCWs amid the COVID-19 pandemic over a year. The present study suggests that healthcare administrators and governments may need to have structured disaster management plans and support systems to mitigate the burnout of HCWs. Systematic scoping review or meta-analysis of the existing literature, or multi-center prospective cohort studies at the international level are warranted as models for future research to establish factors related to burnout among HCWs during the COVID-19 pandemic or other health emergencies.

## 5. Conclusions

Through the study, we have summarized the prevalence of burnout among HCWs during the COVID-19 pandemic over time and showed that HCWs might suffer from burnout considerably during the surges of COVID-19 cases than post-surge periods. Given the uncertainty concerning the pandemic, people are becoming more frustrated with restrictions and mandates, which has turned into stigmatization and hatred against HCWs. Exhaustion from the care of COVID-19 patients, rationing of care amid the surge of COVID-19 cases and healthcare overwhelm, and attacks against HCWs, all lead to further burnout of HCWs. In addition to personal efforts to prevent burnout, high-level interventions and support for HCWs at the government level may be essential to overcome the prolonged public health emergency.

## Figures and Tables

**Table 1 ijerph-18-11581-t001:** Demographic characteristics of the study participants.

Characteristic	Period 1 11/2020			Period 2 2/2021			Period 3 5/2021		
	Value	SD	95% CI	Value	SD	95% CI	Value	SD	95% CI
Years in experience									
Mean	11.5	8.2	8.6–14.4	11.0	6.9	8.7–13.3	11.0	8.2	8.8–13.1
Gender, no. (%)									
Female	24 (72.7)			24 (66.7)			42 (75.0)		
Male	9 (27.3)			12 (33.3)			14 (25.0)		
Affiliation, no. (%)									
Emergency department/Intensive care unit	8 (24.2)			17 (47.2)			10 (17.9)		
General Medicine	24 (72.7)			15 (41.7)			27 (48.2)		
Others	1 (0.3)			4 (11.1)			19 (34.0)		
Job category, no. (%)									
Physician	11 (33.3)			17 (47.2)			14 (25.0)		
Nurse	21 (63.6)			13 (36.1)			39 (69.6)		
Pharmacist	0			2 (5.6)			0		
Clinical engineer	1 (0.3)			2 (5.6)			0		
Physical therapist	0			2 (5.6)			3 (5.4)		
Size of household									
Mean	2.2	1.1	1.8–2.6	2.5	1.4	2.1–3.0	2.2	1.2	1.8–2.5
Engaged in care of COVID-19 patients or COVID-19 PUI in the past 2 weeks, no. (%)									
No	21 (63.6)			10 (27.8)			5 (8.9)		
Yes	12 (36.4)			26 (72.2)			51 (91.1)		
**Total number of participants**	33			36			56		

Abbreviations: CI, confidence interval; SD, standard deviation; PUI, patients under investigation.

**Table 2 ijerph-18-11581-t002:** Results of the Maslach Burnout Index.

Measure	Period 1 11/2020					Period 2 2/2021					Period 3 5/2021				
	Engaged in COVID-19 Care in the Last 2 Weeks (*n* = 12)		Not Engaged in COVID-19 Care in the Last 2 Weeks (*n* = 21)		*p*-Value	Engaged in COVID-19 Care in the Last 2 Weeks (*n* = 26)		Not Engaged in COVID-19 Care in the Last 2 Weeks (*n* = 10)		*p*-Value	Engaged in COVID-19 Care in the Last 2 Weeks (*n* = 51)		Not Engaged in COVID-19 Care in the Last 2 Weeks (*n* = 5)		*p*-Value
	Median	IQR	Median	IQR		Median	IQR	Median	IQR		Median	IQR	Median	IQR	
EE	24.0	14.3–38.0	18.0	12.5–23.5	0.116	17.0	12.0–28.5	23.0	8.0–30.3	0.915	20.0	12.0–31.0	13.0	6.5–28.0	0.238
DP	4.5	2.3–12.3	4.0	0.5–6.5	0.207	2.5	1.0–6.5	4.0	3.8–5.3	0.255	5.0	1.0–12.0	5.0	1.5–7.5	0.806
PA	27.5	20.0–36.0	20.0	16.5–28.5	0.088	24.5	18.8–32.3	23.0	15.0–25.5	0.313	25.0	19.0–30.0	15.0	11.0–34.5	0.446
Measure (No. (%))															
Burnout					0.016					0.964					0.295
Yes	6 (50.0)		2 (9.5)			8 (30.8)		3 (30.0)			22 (43.1)		1 (20.0)		
EE ≥ 27	3/6 (50.0)		2/2 (100)			7/8 (87.5)		3/3 (100)			19/22 (86.4)		1/1 (100)		
DP ≥ 10	4/6 (66.7)		1/2 (50.0)			4/8 (50.0)		0/3 (0)			15/22 (68.2)		1/1 (100)		
No	6 (50.0)		19 (90.5)			18 (69.2)		7 (70.0)			29 (56.9)		4 (80.0)		

Abbreviations: CI, confidence interval. The base date of the “last 2 weeks” is the day when the participants answered the surveys. *p*-value is calculated with Mann–Whitney U test or Fisher’s exact test.

**Table 3 ijerph-18-11581-t003:** Prevalence of burnout depending on variables based on logistic regression analyses.

Variable	Period 1 11/2020		Period 2 2/2021		Period 3 5/2021	
	OR (95% CI)	*p*-Value	OR (95% CI)	*p*-Value	OR (95% CI)	*p*-Value
**Gender**						
**Male**	(Reference)	NA	(Reference)	NA	(Reference)	NA
**Female**	0.53 (0.10–2.8)	0.651	2.1 (0.49–9.4)	0.446	1.35 (0.39–4.7)	0.639
**Job category**						
**Physician**	(Reference)	NA	(Reference)	NA	(Reference)	NA
**Nurse**	1.2 (0.23–6.3)	0.830	0.43 (0.086–2.1)	0.440	1.5 (0.44–5.4)	0.501
**Female × Nurse ***	NA	NA	NA	NA	2.4 (0.23–25.5)	0.460
**Affiliation**						
**General Medicine**	(Reference)	NA	(Reference)	NA	(Reference)	NA
**EICU**	6.7 (1.1–42.1)	0.031	0.62 (0.13–2.9)	0.699	1.9 (0.43–8.2)	0.404
**EICU × Female**	12.0 (1.2–121.6)	0.035	0.83 (0.051–13.6)	0.898	1.0 (0.16–6.2)	1.00
**Engagement in COVID-19 care** **in the last 2 weeks**						
**No**	(Reference)	NA	(Reference)	NA	(Reference)	NA
**Yes**	8.5 (1.3–54.1)	0.014	1.0 (0.21–5.1)	1.00	3.0 (0.32–29.1)	0.394
**Yes × Female**	12.8 (1.3–128.8)	0.031	2.0 (0.19–20.6)	0.560	2.4 (0.23–25.5)	0.460

Abbreviations: CI, confidence interval; NA, not applicable; OR, odds ratio; EICU, emergency intensive care unit. The base date of the “last 2 weeks” is the day when the participants answered the survey. * The impact of gender in terms of job category and burnout were not assessed in period 1 and 2 due to the limited sample sizes.

## Data Availability

The datasets generated and analyzed during the current study are available from the corresponding author on reasonable request.

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
