# Peer review of "Burnout of Healthcare Workers Amid the COVID-19 Pandemic: A Follow-Up Study"

_ijerph, 2021, doi:10.3390/ijerph182111581_

Round 1

Reviewer 1 Report

It is an interesting article as it looks at the level of burnout in professionals during the global pandemic.
As they provide in the study, it would have been better to generalise the results in a larger hospital or health centre, which served a larger population. This could be a line to continue this research, in which the levels of stress among professionals before the pandemic could be collected, and to analyse whether it is caused by the pandemic situation, or by the scarce material and human resources and the poor organisation of the centres.

Author Response

  • Comment 1: [It is an interesting article as it looks at the level of burnout in professionals during the global pandemic. As they provide in the study, it would have been better to generalise the results in a larger hospital or health centre, which served a larger population. This could be a line to continue this research, in which the levels of stress among professionals before the pandemic could be collected, and to analyse whether it is caused by the pandemic situation, or by the scarce material and human resources and the poor organisation of the centres.]

Response: We truly appreciate you for reviewing our manuscript. We believe it is crucial to report timely data during the pandemic. At the same time, as suggested by you, we are planning to continue and further expand the study to a national or global level through a network of academic society to see the generalizability of the current results and associated factors of burnout in healthcare workers.

Reviewer 2 Report

The topic is very relevant and researches in this scientific field are necessary. This article analyses the newest research in the field of burnout but their article is not a systematic literature review and the practical or scientific value is not created from systemic analyses of literature.

The authors correctly note in limitations that their research is made with a small sample size and even more just in one organization. Thus research value is low even if the researchers applied great methods for data analyses. I would suggest expanding the research by either making a systematic literature review, either exploring a reliable sample size of medical organizations.

Author Response

  • Comment 1: [The topic is very relevant and researches in this scientific field are necessary. This article analyses the newest research in the field of burnout but their article is not a systematic literature review and the practical or scientific value is not created from systemic analyses of literature. The authors correctly note in limitations that their research is made with a small sample size and even more just in one organization. Thus research value is low even if the researchers applied great methods for data analyses. I would suggest expanding the research by either making a systematic literature review, either exploring a reliable sample size of medical organizations.]

Response: Thank you very much for reviewing our manuscript. As noted, a small sample size with single-center data is certainly a limitation of the study we mentioned in the discussion. Despite limitations, we believe it would be worth reporting the data and the implications given the extent of the COVID-19 pandemic and the importance of reporting the results in a timely manner. To address the limitations, we are planning to continue and further expand the study to a national or global level using networks through academic societies. As suggested, we would proceed with a scoping review or a meta-analysis to bring up more robust evidence.

Reviewer 3 Report

Minor language editing is needed.

The study is well-designed when it comes to methodology, with an appropriate, validated tool to measure the extent of burnout. However, for a study of such a methodolgy, the response rate and the eventual study group is strikingly low. It should not be difficult to reach higher numbers of responses: it would be expected to collect a numer of responses closer to 100 than just 33-36 in the initial phases. With low response rate, perhaps a larger numer of HCW should have been targeted. Also, as outlined in the discussion, the follow-up phases also comprise different samples. I have my reservations whether the results derived from such a small sample will significantly improve our knowledge on the burnout in HCW during the COVID-19 pandemic.

Author Response

  • Comment 1: [Minor language editing is needed. The study is well-designed when it comes to methodology, with an appropriate, validated tool to measure the extent of burnout. However, for a study of such a methodolgy, the response rate and the eventual study group is strikingly low. It should not be difficult to reach higher numbers of responses: it would be expected to collect a numer of responses closer to 100 than just 33-36 in the initial phases. With low response rate, perhaps a larger numer of HCW should have been targeted. Also, as outlined in the discussion, the follow-up phases also comprise different samples. I have my reservations whether the results derived from such a small sample will significantly improve our knowledge on the burnout in HCW during the COVID-19 pandemic.]

Response: We appreciate your time for reviewing our manuscript. While it was unfortunate that we had a low response rate, previous studies noted that web-based surveys on burnout without financial incentives consistently had response rates of 25.3-32%1-3. Thus, our response rate would be considered similar to previous studies. However, we certainly acknowledge the limitation as noted as a limitation in the discussion section. Despite the limitation, we believe it would be worth reporting the data and the implications given the extent of the COVID-19 pandemic and the importance of reporting the results in a timely manner. Also, an experienced scholarly writer (English native) has edited this manuscript.

Round 2

Reviewer 2 Report

The authors improved the article by adding deeper analyses of literature review. The research gives value because it includes tree periods of Covid-19 analyses about burnout in hospital. Limitation as noted by authors remains analyses juts one hospital situation. The article could be improved determining some models and perspectives for future research, and describing the current research value. The article could be published after minor revision.

Author Response

  • Comment 1: [The authors improved the article by adding deeper analyses of literature review. The research gives value because it includes tree periods of Covid-19 analyses about burnout in hospital. Limitation as noted by authors remains analyses juts one hospital situation. The article could be improved determining some models and perspectives for future research, and describing the current research value. The article could be published after minor revision.]

Response: Thank you very much for reviewing our manuscript again. The value of the present study is noted in the discussion section (line 219 to 221). Also, we have described directions for future studies (line 221 to 225).